# Thermoresponsive Ionic Liquid with Different Cation–Anion Pairs as Draw Solutes in Forward Osmosis

**DOI:** 10.3390/molecules27248869

**Published:** 2022-12-13

**Authors:** DaEun Yang, Hyo Kang

**Affiliations:** BK-21 Four Graduate Program, Department of Chemical Engineering, Dong-A University, Busan 49315, Republic of Korea

**Keywords:** forward osmosis, draw solute, phosphonium and ammonium derivatives, thermoresponsive ionic liquid

## Abstract

We synthesized various phosphonium- and ammonium-based ionic liquids (ILs), using benzenesulfonate (BS) and 4-methylbenzenesulfonate (MBS) to establish the criteria for designing an ideal draw solute in a forward osmosis (FO) system. Additionally, the effects of monocationic, dicationic, and anionic species on FO performance were studied. Monocationic compounds ([P_4444_][BS], [P_4444_][MBS], [N_4444_][BS], and [N_4444_][MBS]) were obtained in one step via anion exchange. Dicationic compounds ([(P_4444_)_2_][BS], [(P_4444_)_2_][MBS], [(N_4444_)_2_][BS], and [(N_4444_)_2_][MBS]) were prepared in two steps via a Menshutkin SN_2_ reaction and anion exchange. We also investigated the suitability of ILs as draw solutes for FO systems. The aqueous [P_4444_][BS], [N_4444_][BS], [N_4444_][MBS], and [(N_4444_)_2_][BS] solutions did not exhibit thermoresponsive behavior. However, 20 wt% [P_4444_][MBS], [(P_4444_)_2_][BS], [(P_4444_)_2_][MBS], and [(N_4444_)_2_][MBS] had critical temperatures of approximately 43, 33, 22, and 60 °C, respectively, enabling their recovery using temperature. An increase in IL hydrophobicity and bulkiness reduces its miscibility with water, demonstrating that it can be used to tune its thermoresponsive properties. Moreover, the FO performance of 20 wt% aqueous [(P_4444_)_2_][MBS] solution was tested for water flux and found to be approximately 10.58 LMH with the active layer facing the draw solution mode and 9.40 LMH with the active layer facing the feed solution.

## 1. Introduction

The scarcity of clean water is recognized as a serious global environmental problem [1,2,3]. Various wastewater treatment systems have been considered to overcome water shortages and pollution [4,5]. Forward osmosis (FO) is a membrane separation system for wastewater treatment [6,7,8]. In the FO process, water diffusion occurs from a low-concentration solution (feed) to a high-concentration solution (draw) until osmotic equilibrium is achieved. The water flow in this process is driven by a difference in osmotic pressure between the two solutions [9]. The ability of the draw solute to draw water is vital for obtaining a high water flux [10,11,12,13,14]. After the permeation of water, the diluted draw solution and draw solute can be separated for water recovery and draw solute regeneration. Although the study of draw solutes in FO systems is of great interest, the recovery of draw solutes remains a challenging topic [15,16]. For an FO system to work, a dehydration process should be performed to separate the draw solute and water from the diluted draw solution. To increase energy efficiency, reducing the energy required for this process is important. To date, various draw solutes have been proposed, including inorganic salts, responsive polymers, hydrophilic polymers, thermoresponsive hydrogels, stimuli-responsive nanoparticles, and thermoresponsive organic compounds [17,18,19,20,21,22,23,24,25]. Among the proposed draw solutes, thermoresponsive organic compounds can be realized as desirable draw solutes with high osmotic pressure and easy recovery and reuse [26,27,28].

Ionic liquids (ILs), organic salts in the liquid state that can generate high osmotic pressures, are interesting as draw solutes. Appropriate ILs can be designed because the properties of ILs can be tuned with various combinations of cations and anions [29,30]. Furthermore, the synthesis conditions of ILs with thermoresponsive behavior can be controlled to obtain a hydrophilic/hydrophobic balance. Therefore, a more suitable IL-based draw solute can be developed by controlling the molecular structure of IL. Variations of some key parameters affecting the structure can be introduced, such as the nature of the anion/cation and the alkyl chain length at the anion/cation [31,32,33,34,35]. Most studies on ILs have focused on monocationic-based ILs, but recently, dicationic-based ILs have been studied because of their unique properties, such as remarkable thermal stability and surface tension [36,37,38,39]. Consequently, both monocationic-based ILs and dicationic-based ILs are attractive research targets in the fields of physics and chemistry [40,41]. This indicates that properly designed monocationic-based ILs and dicationic-based ILs achieve the desired physicochemical and biological properties. The selection of an appropriate counter ion is also very important for IL characterization [42]. Representative anions of ILs include sulfonate, carboxylate, phosphate, halides, and amino acids [43,44,45]. These anions are used to control water miscibility by altering the alkyl chain length or by attaching non-polar groups to maintain a hydrophilic/hydrophobic balance [46,47,48]. For this reason, it is necessary to pay attention to benzenesulfonate, which is a form in which phenyl, a simple hydrophobic group, is attached to a sulfonate group. According to the previous literature, benzenesulfonate is the simplest member of the class of a benzenesulfonic acids, with high thermal stability and low manufacturing costs [49,50].

Thermoresponsive ILs have been used as draw solutes for FO because they exhibit a change in water solubility with temperature and can be broadly classified into two types [51,52,53]. In upper critical solution temperature (UCST) types, the IL and water are immiscible below the critical temperature; when the mixture is heated above the critical temperature, it becomes a miscible and homogeneous phase. In lower critical solution temperature (LCST) types, the homogeneous phase of the IL/water mixture turns into a separated phase when heated above the critical temperature. An advantage of thermoresponsive ILs, especially the LCST-type, is that they can be dehydrated by slight temperature changes when diluted in the FO process [54,55,56,57]. In addition, the energy efficiency can be improved by recovering the draw solutes using thermoresponsive draw solutes that exhibit UCSTs or LCSTs similar to the operating temperature of the FO process [58,59].

In this study, phosphonium-based and ammonium-based ILs were synthesized using benzenesulfonate (BS) and 4-methylbenzenesulfonate (MBS) to provide design ideas for IL as draw solutes. In addition, we performed a systematic analysis of the effects of monocationic, dicationic, and anionic compounds on their thermoresponsive behavior and FO performance.

## 2. Results and Discussion

### 2.1. Synthesis of Monocationic and Dicationic ILs

Figure 1 shows the chemical structure of (a) the cations and (b) the BS and MBS anions of the ILs analyzed in this study. This series shows the effect of structural changes, such as deformation in the monocationic and dicationic forms, the central atom in the cation (phosphonium or ammonium), and an increase in the number of methyl groups in the anion. The monocationic and dicationic structures were designed and synthesized according to the simple protocol shown in Figure 2 using modified literature procedures [60,61,62]. Monocationic compounds were obtained in one step via anion exchange, and dicationic compounds were prepared in two steps via the Menshutkin SN_2_ reaction and anion exchange methods.

Structural analysis of the ILs used in this study was performed using ^1^H-NMR spectroscopy. The ^1^H-NMR spectra of the monocationic series are shown in Figure 3a–d. In Figure 3a, the spectrum of [P_4444_][BS] contains peaks from the aromatic group of the benzenesulfonate moiety (δ = 7.48−7.68 (peaks g and f) and 7.68−7.90 ppm (peak e)) and peaks from alkyl groups of the tetrabutylphosphonium moiety (δ = 0.69−0.89 (peak a), 1.18−1.71 (peaks b and c), and 2.08−2.14 ppm (peak d)). The ^1^H-NMR spectrum of [P_4444_][MBS] contains additional peaks corresponding to added methyl protons when compared to the spectrum of [P_4444_][BS]: δ = 1.99−2.38 (peak e, Figure 3b). As shown in Figure 3c, the spectrum of [N_4444_][BS] contains peaks from the aromatic group of the benzenesulfonate moiety (δ = 7.49−7.59 (peaks g and f) and 7.76−7.81 ppm (peak e)) and peaks from the alkyl groups of the tetrabutylammonium moiety (δ = 0.88−0.99 (peak a), 1.18−1.40 (peak b), 1.57−1.68 (peak c), and 3.09−3.22 ppm (peak d)). The ^1^H-NMR spectrum of [N_4444_][MBS] contains additional peaks corresponding to the added methyl protons when compared to the spectrum of [N_4444_][BS]: δ = 3.06−3.21 (peak e, Figure 3d).

The ^1^H-NMR spectra of the dicationic series are shown in Figure 3e–h. In Figure 3e, the spectrum of [(P_4444_)_2_][BS] contains peaks from the aromatic group of the benzenesulfonate moiety (δ = 7.50−7.60 (peak d), 7.77−7.84 ppm (peak e)) and alkyl peaks in the tetrabutylphosphonium moiety (δ = 0.89–0.99 (peak a), 1.30–1.64 (peak b), and 2.09–2.22 (peak c)). The ^1^H-NMR spectrum of [(P_4444_)_2_][MBS] contains additional peaks corresponding to the added methyl protons when compared to the spectrum of [(P_4444_)_2_][BS]: δ = 2.33−2.43 (peak d, Figure 3f). In Figure 3g, the spectrum of [(N_4444_)_2_][BS] contains peaks from the aromatic group of the benzenesulfonate moiety (δ = 7.78−7.83 (peak e), 7.52−7.63 ppm (peak f)) and the tetrabutylammonium moiety (δ = 0.83–1.00 (peak a), 1.19–1.41 (peak b), 1.54–1.71 (peak c), and 3.11–3.26 (peak d)). The ^1^H-NMR spectrum of [(N_4444_)_2_][MBS] contains additional peaks corresponding to the added methyl protons when compared to the spectrum of [(N_4444_)_2_][BS]: δ = 2.36−2.43 (peak d, Figure 3h). In all the ILs, the integrated area of each region matches the predicted proportion of hydrogen atoms. Moreover, anion exchange from [(P_4444_)_2_][Br] to [(P_4444_)_2_][MBS] was confirmed using FT-IR spectroscopy, as shown in Figure 4. The absorption of [(P_4444_)_2_][MBS] from 2968 to 2930 cm^−1^ corresponds to the butyl groups in [(P_4444_)_2_][Br]. Therefore, this region can be used to identify changes in the cations. The asymmetric stretching vibration peak of SO_3_^−^ located at 1380–1150 cm^−1^, and the symmetric stretching peaks of SO_3_^−^ located at 1100–1010 cm^−1^ confirm changes in the anions. In other words, the peaks corresponding to the C-H and S=O groups allow the interpretation of the IL structure. Therefore, we conclude that both monocationic and dicationic ILs were successfully synthesized.

### 2.2. Conductivity

The conductivity of ILs originates from the intrinsic motion of cations and anions, affected by the degree of ionic dissociation, and is related to the driving force of the FO process [63]. We investigated the conductivity of the monocationic and dicationic ILs to determine the solute ion number and ion mobility in the draw solutions. Figure 5 shows the conductivities of aqueous [P_4444_][BS], [P_4444_][MBS], [N_4444_][BS], [N_4444_][MBS], [(P_4444_)_2_][BS], [(P_4444_)_2_][MBS], [(N_4444_)_2_][BS], and [(N_4444_)_2_][MBS] solutions at concentrations of 20, 15, 10, and 5 wt% at room temperature. As the concentration of all measured draw solutions increased, the conductivity increased owing to its intrinsic properties. However, it is known that in a high concentration range outside of a concentration range in which conductivity and concentration are proportional, the interaction force between ions is strengthened and the mobility of ions is weakened, thereby decreasing the conductivity of the aqueous solution [64]. Eight ILs with different cation and anion combinations were investigated. The conductivity of dicationic ILs was higher than that of monocationic ILs when using the same anion and cation central atom. These results are explained by the fact that the conductivity is affected by the number of ions [65,66]. That is, the conductivity of a dicationic IL is higher than that of the monocationic IL because a dicationic IL has more IL moieties in an aqueous solution than a monocationic IL. In addition, the conductivity of ILs with [N]^+^ and/or [BS]^−^ was higher than that of ILs with [P]^+^ and/or [MBS]^−^ at all solution concentrations tested. This can be explained by hydrophilicity and size effects [67,68]. Increased hydrophilicity and decreasing ionic size of ILs lead to higher mobility, and hence, higher conductivity. ILs with [N]^+^ and/or [BS]^−^ are more hydrophilic and contain smaller ions than ILs with [P]^+^ and/or [MBS]^−^, respectively. Therefore, ILs with small, hydrophilic ions, such as [N]^+^ and/or [BS]^−^, induce relatively high conductivity.

### 2.3. Osmotic Pressure

Osmosis is a natural phenomenon that causes water to diffuse from a feed solution (low osmotic pressure) to a draw solution (high osmotic pressure). In the FO process, the draw solute generates a high osmotic pressure in the aqueous solution for water diffusion from the feed solution to the draw solution, which is related to the FO performance [69]. The osmotic pressure (*Π*) is mainly described by the van ‘t Hoff equation (Equation (1)). It is a function of the solution concentration [70].
*Π* = *C_i_RT*(1)

Here, *C_i_* is the molar concentration of solute *i*, *R* is the ideal gas constant, and *T* is the temperature.

As shown in Figure 6, because the osmotic pressure difference between the feed and draw solutions is the driving force of the FO process, osmotic pressure can be measured using freezing point depression to investigate the possible applications of an IL as a draw solute. As the concentration of all the draw solutions increased, the osmotic pressure increased. This result proves the validity of the van’t Hoff equation (Equation (1)). That is, according to Equation (1), the proportional relationship between the osmotic pressure and the concentration can be applied even to the highest concentration at which the draw solute can be dissolved in water. The osmotic pressure results for the monocationic ILs are shown in Figure 6a, and Figure 6b shows the osmotic pressure results for the dicationic ILs. Osmotic pressure is known to be affected by hydrophilic ionic groups [71]. Therefore, the fact that osmotic pressure increased with increasing the number of ionic moieties can explain the higher osmotic pressure of dicationic ILs than that of monocationic ILs at all solution concentrations tested. In addition, osmotic pressure is correlated with the van ‘t Hoff coefficient, which affects the solubility of the solute [72]. For example, ILs with [N]^+^ and/or [BS]^−^ have a higher solubility than ILs with [P]^+^ and/or [MBS]^−^ because of the relatively high polarity and small size of the ionic moieties. Thus, the osmotic pressure of ILs with [N]^+^ and/or [BS]^−^ is higher than that of ILs with [P]^+^ and/or [MBS]^−^. In addition to solubility, molecular weight plays an important role in generating osmotic pressure, which means that decreasing the molecular weight increases the osmotic pressure [73]. Therefore, the osmotic pressure of ILs can also be described as a well-known colligative property that depends on the concentration of the solute. Specifically, although the weight concentrations were the same, the molecular weights of ILs with [P]^+^ and/or [MBS]^−^ are greater than those of ILs with [N]^+^ and/or [BS]^−^. Thus, the osmotic pressure is higher because the molar concentration of ILs with [N]^+^ and/or [BS]^−^ is higher than that of ILs with [P]^+^ and/or [MBS]^−^ in all concentration ranges.

### 2.4. Thermoresponsive Behavior

In the FO process, it is important to ensure that the diluted thermoresponsive draw solution can be reused as a draw solute using a temperature system [74]. By measuring the LCST, it was confirmed that the draw solute could be reused in the regeneration process for separating the diluted draw solution and water after the FO process. The critical temperature at which the aqueous ILs phase separates was determined by measuring the transmittance of these solutions at a constant wavelength (650 nm) with respect to the temperature. When the temperature decreases below the critical temperature of the draw solution, molecular motion decreases, forming hydrogen bonds between the water molecules and the ionic groups of the draw solute, resulting in complete mixing [75]. Therefore, below the LCST, the transmittance of the aqueous solution is close to 100% because the draw solution is optically transparent and homogeneous. As the temperature increases above the critical temperature, the increase in molecular motion breaks the hydrogen bonds between the water molecules and the ionic groups of the draw solute, resulting in IL aggregation as ion–ion interactions become dominant [76,77]. Therefore, above the LCST, the aqueous solution becomes cloudy and blocks light, so the transmittance of the aqueous solution was approximately 0% [78]. The LCST is the change in the draw solute/water mixture from a homogeneous (high transmittance) to a heterogeneous (low transmittance) state with an increase in temperature. The [P_4444_][BS], [N_4444_][BS], [N_4444_][MBS], and [(N_4444_)_2_][BS] aqueous solutions did not show any changes in transmittance according to the temperature change, indicating no LCST. In contrast, aqueous solutions of [P_4444_][MBS], [(P_4444_)_2_][BS], [(P_4444_)_2_][MBS], and [(N_4444_)_2_][MBS] were found to have critical temperatures, as shown in Figure 7. They exhibited a typical LCST-type phase separation behavior by changing the IL concentration as shown in their phase diagrams (Figure 7e). The LCST behavior of monocationic and dicationic-based ILs can explain by two effects: (1) a hydrophobic effect and (2) a steric effect [79,80]. An increase in IL hydrophobicity and bulkiness reduces its miscibility with water, demonstrating that it can be used to tune its thermoresponsive properties. [MBS]^−^-based ILs are more hydrophobic than [BS]^−^-based ones. For example, [P_4444_][BS] and [(N_4444_)_2_][BS] did not exhibit LCST behavior, whereas [P_4444_][MBS] and [(N_4444_)_2_][MBS] showed LCST behavior. In addition, [P]^-^-based ILs are more hydrophobic than [N]^−^-based ones. For example, [N_4444_][MBS] and [(N_4444_)_2_][BS] did not exhibit LCST behavior, but [P_4444_][MBS] and [(P_4444_)_2_][BS] showed LCST behavior. Because of the steric effect, the dicationic moiety induces more IL aggregation and causes the IL to form a heterogeneous phase with water. For example, [P_4444_][BS] and [N_4444_][MBS] did not exhibit LCST behavior, whereas [(P_4444_)_2_][BS] and [(N_4444_)_2_][MBS] showed LCST behavior. When the temperatures of the aqueous [P_4444_][MBS], [(P_4444_)_2_][BS], [(P_4444_)_2_][MBS], and [(N_4444_)_2_][MBS] solutions were lower than the LCST, the hydrogen bond interactions with water molecules were stronger than the ion–ion interactions between the phosphonium/ammonium cations and the BS and MBS anions. Thus, [P_4444_][MBS], [(P_4444_)_2_][BS], [(P_4444_)_2_][MBS], or [(N_4444_)_2_][MBS] and water formed homogeneous phases. On the other hand, increasing the temperature above the LCST causes ion–ion interactions to become stronger than the hydrogen bond interactions with water molecules, resulting in IL aggregation. In addition, at 15 wt%, the LCSTs of [P_4444_][MBS], [(P_4444_)_2_][BS], [(P_4444_)_2_][MBS], and [(N_4444_)_2_][MBS] were 44, 35, 28, and 61 °C, respectively. At 20 wt%, however, the LCSTs decreased to 43, 33, 22, and 60 °C, respectively. This observation suggests that an increase in the concentration of the aqueous solution results in a lower LCST. However, it is known that a reversed T_cloud_/concentration relationship is observed at much higher draw solution concentrations [81,82,83]. LCSTs within the range of 20–25 °C have been reported to be energy efficient for draw solute recovery, suggesting that [(P_4444_)_2_][MBS] is a promising draw solute for FO systems [84]. Therefore, [(P_4444_)_2_][MBS] can be separated from water by simply adjusting its solubility by varying the temperature. The thermal energy required for this can be obtained from geothermal heat and/or waste heat from industrial processes [85].

### 2.5. Water Flux

In the FO process, a higher water flux corresponds to a larger volume of water passing through the FO membrane per unit time [86]. Therefore, water flux should be considered while investigating the effect of draw solutes on FO performance. The higher osmotic pressure of the draw solution than that of feed solution can create a driving force for water transport from the feed solution, resulting in a higher water flux to the draw solution in the FO process [87]. Thus, the order of the flux results for the eight draw solutes used in this study is expected to be the same as the previous osmotic pressure results (Figure 6). Based on previous experimental results, among these eight draw solutes, [(P_4444_)_2_][MBS] was chosen as a representative IL because it exhibits relatively good FO performance and the most energy efficient LCST. Its water flux was measured at four concentrations (5, 10, 15, and 20 wt%), in two different operational modes: (1) AL-FS, where the active layer faced the feed solution, and (2) AL-DS, where the active layer faced the draw solution. The connected glass tubes were filled with DI water on one side and [(P_4444_)_2_][MBS] solution on the other side. Subsequently, a thin-film composite membrane was placed between the connected tubes, and measurements were performed below the LCST of the [(P_4444_)_2_][MBS] aqueous solution. The results are shown in Figure 8. The concentration of the draw solution has a key effect on FO performance because a higher concentration in the draw solution induces a higher osmotic pressure, thereby improving water flux in the FO system. As expected, the water flux of [(P_4444_)_2_][MBS] increased as the concentration increased. For example, the water flux was approximately 5.92 (5 wt%), 6.97 (10 wt%), 8.90 (15 wt%), and 10.58 (20 wt%) LMH in the AL-DS mode and approximately 4.35 (5 wt%), 4.64 (10 wt%), 5.40 (15 wt%), and 9.40 (20 wt%) LMH in the AL-FS mode. In addition, the water flux varied with the orientation of the membrane (FO operating conditions) [88]. In the AL-FS mode, water molecules penetrate the active layer of the feed solution and dilute the draw solution in the porous layer, thereby reducing the effective osmotic driving force. This dilutive internal concentration polarization degrades membrane performance. Therefore, compared with the AL-FS mode, the AL-DS mode tends to have a higher FO water flux.

### 2.6. Recyclability Study of [(P_4444_)_2_][MBS]

To demonstrate the recyclability of [(P_4444_)_2_][MBS] after osmotic experiments, FO process was repeated four times using a 20 wt% solution of [(P_4444_)_2_][MBS] as the draw solution and DI water as the feed solution. When the temperature rises above the critical temperature after the permeation process, [(P_4444_)_2_][MBS] is precipitated in the solution, and pure water can be easily separated by a simple filtration process. As shown in Figure 9, to confirm the recyclability of [(P_4444_)_2_][MBS], the osmotic pressure and thermoresponsive behavior tests of [(P_4444_)_2_][MBS] were measured at the fourth run, respectively. Osmotic pressure and LCST value of [(P_4444_)_2_][MBS] in four runs are almost similar to the pristine [(P_4444_)_2_][MBS]. These recycling results clearly show that [(P_4444_)_2_][MBS] can be easily recycled with relatively low energy consumption without significant loss.

## 3. Experimental

### 3.1. Reagents

Sodium benzenesulfonate ([Na][BS]), sodium 4-methylbenzenesulfonate ([Na][MBS]), tetrabutylphosphonium bromide, tetrabutylammonium bromide, and 1,8-dibromooctane were purchased from Tokyo Chemical Industry (TCI) Co., Ltd. (Tokyo, Japan) Tributylamine, tributylphosphine, and dichloromethane were purchased from Sigma–Aldrich Co.(Saint Louis, MO, USA), LLC. Ethyl ether, ethyl acetate, acetone, and nitric acid were purchased from Daejung Chemicals and Metals Co., Ltd. (Sinan, Republic of Korea) Silver nitrate was obtained from Junsei Chemical Co., Ltd. (Tokyo, Japan) All the other chemicals were purchased without further purification.

### 3.2. Synthesis of Monocationic and Dicationic ILs

#### 3.2.1. Monocationic Phosphonium-Based ILs

Tetrabutylphosphonium benzenesulfonate ([P_4444_][BS]) was obtained by stirring tetrabutylphosphonium bromide (4.00 g, 10 mmol) and sodium benzenesulfonate (3.59 g, 20 mmol) in DI water (30 mL) in a glass bottle. After the reaction was completed at room temperature for 24 h, the resulting solution was slowly dripped into 100 mL of dichloromethane. The resulting solid was then washed three times with DI water until it passed the AgNO_3_ test, commonly known as halide analysis of IL [89]. It was then dried at approximately 100 °C to obtain the final product. The yield of the product was above 80%. ^1^H-NMR [400 MHz, D_2_O]: δ 0.69−0.89 (t, 12H, -(CH_2_-CH_2_-CH_2_-*CH*_3_)_4_), 1.18−1.71 (m, 16H, -(CH_2_-*CH*_2_*-CH*_2_-CH_3_)_4_)), 2.08−2.14 (m, 8H, -(*CH*_2_-CH_2_-CH_2_-CH_3_)_4_), 7.48−7.68 (s, 3H, *PhH*-), 7.68−7.90 (s, 2H, *PhH*-).

Tetrabutylphosphonium 4-methylbenzenesulfonate ([P_4444_][MBS]) was obtained by stirring tetrabutylphosphonium bromide (4.00 g, 10 mmol) and sodium 4-methylbenzenesulfonate (4.16 g, 20 mmol) in DI water (30 mL) in a glass bottle. After the reaction was completed at room temperature for 24 h, the resulting solution was slowly dripped into 100 mL of dichloromethane. The product was then washed three times with DI water until it passed the AgNO_3_ test and dried at approximately 100 °C to obtain the final product. The yield of the product was above 80%. ^1^H-NMR [400 MHz, CDCl_3_]: δ 0.79−1.12 (t, 12H, -(CH_2_-CH_2_-CH_2_-*CH*_3_)_4_), 1.42−1.71 (m, 16H, -(CH_2_-*CH*_2_*-CH*_2_-CH_3_)_4_), 1.99−2.38 (m, 8H, -(*CH*_2_-CH_2_-CH_2_-CH_3_)_4_, s, 3H, *CH*_3_-Ph-), 7.00−7.14 (s, 2H, CH_3_-*PhH*-), 7.71−7.83 (s, 2H, CH_3_-*PhH*-).

#### 3.2.2. Monocationic Ammonium-Based ILs

Tetrabutylammonium benzenesulfonate ([N_4444_][BS]) was obtained by stirring tetrabutylammonium bromide (3.23 g, 10 mmol) and sodium benzenesulfonate (3.59 g, 20 mmol) in DI water (30 mL) in a glass bottle. After the reaction was completed at room temperature for 24 h, the resulting solution was slowly dripped into 100 mL of dichloromethane. Finally, the product was washed three times with DI water until it passed the AgNO_3_ test and dried at approximately 100 °C to obtain the final product. The yield of the product was 40–50%. ^1^H-NMR [400 MHz, D_2_O]: δ 0.88−0.99 (t, 12H, -(CH_2_-CH_2_-CH_2_-*CH*_3_)_4_), 1.18−1.40 (m, 8H, -(CH_2_-CH_2_*-CH*_2_-CH_3_)_4_)), 1.57−1.68 (m, 8H, -(CH_2_-*CH*_2_-CH_2_-CH_3_)_4_)), 3.09−3.22 (m, 8H, -(*CH*_2_-CH_2_-CH_2_-CH_3_)_4_), 7.49−7.59 (s, 3H, *PhH*-), 7.76−7.81 (s, 2H, *PhH*-).

Tetrabutylammonium 4-methylbenzenesulfonate ([N_4444_][MBS]) was obtained by stirring tetrabutylammonium bromide (3.23 g, 10 mmol) and sodium 4-methylbenzenesulfonate (4.16 g, 20 mmol) in DI water (30 mL) in a glass bottle. After the reaction was completed at room temperature for 24 h, the resulting solution was slowly dripped into 100 mL of dichloromethane. Finally, the product was washed three times with DI water until it passed the AgNO_3_ test and dried at approximately 100 °C to obtain the final product. The yield of the product was 40–50%. ^1^H-NMR [400 MHz, D_2_O]: δ 0.85−1.00 (t, 12H, -(CH_2_-CH_2_-CH_2_-*CH*_3_)_4_), 1.26−1.42 (m, 8H, -(CH_2_-CH_2_*-CH*_2_-CH_3_)_4_), 1.52−1.68 (m, 8H, -(CH_2_-*CH*_2_-CH_2_-CH_3_)_4_), 2.34−2.40 (m, 8H, -(*CH*_2_-CH_2_-CH_2_-CH_3_)_4_), 3.06−3.21 (s, 3H, *CH*_3_-Ph-), 7.29−7.37 (s, 2H, CH_3_-*PhH*-), 7.63−7.70 (s, 2H, CH_3_-*PhH*-).

#### 3.2.3. Dicationic Phosphonium-Based ILs

In this study, octane-1,8-diylbis(tributylphosphonium) dibromide ([(P_4444_)_2_][Br]) was first obtained by stirring tributylphosphine (20.0 g, 100 mmol) and 1,8-dibromooctane (13.60 g, 50 mmol) in acetone (40 mL) in a glass bottle. The mixture was stirred at 40 °C for 48 h and then precipitated into 600 mL diethyl ether. The product was then washed several times with diethyl ether, and dried at approximately 100 °C to obtain the final product. The yield of the product was above 80%. ^1^H-NMR [400 MHz, D_2_O]: δ 0.86–0.99 (m, 18H, (*CH_3_*-CH_2_-)), 1.30–1.65 (m, 36H, (-*CH_2_-CH_2_*-CH_2_-P^+^-), (P^+^-CH_2_-*(CH_2_)_6_*-CH_2_-P^+^)), 2.08–2.25 (m, 16H, (-*CH_2_*-P^+^-)).

Octane-1,8-diylbis(tributylphosphonium) benzenesulfonate ([(P_4444_)_2_][BS]) was obtained by stirring [(P_4444_)_2_][Br] (2.50 g, 3.70 mmol) and sodium benzenesulfonate (2.00 g, 11.10 mmol) in DI water (100 mL) in a glass bottle. After stirring the mixture at room temperature for 12 h, the crude product was extracted. It was washed several times with ethyl acetate until it passed the AgNO_3_ test, then dried at approximately 100 °C to obtain the final product. The yield of the product was above 80%. ^1^H-NMR [400 MHz, D_2_O]: δ 0.89–0.99 (m, 18H, (*CH_3_*-CH_2_-)), 1.30–1.64 (m, 36H, (-*CH_2_-CH_2_*-CH_2_-P^+^-), (P^+^-CH_2_-*(CH_2_)_6_*-CH_2_-P^+^)), 2.09–2.22 (m, 16H, (-*CH_2_*-P^+^-)), 7.50−7.60 (s, 6H, *PhH*-), 7.77−7.84 (s, 4H, *PhH*-).

Octane-1,8-diylbis(tributylphosphonium) 4-methylbenzenesulfonate ([(P_4444_)_2_][MBS]) was obtained by stirring [(P_4444_)_2_][Br] (2.50 g, 3.70 mmol) and sodium 4-methylbenzenesulfonate (2.15 g, 11.10 mmol) in DI water (100 mL) in a glass bottle. After stirring the mixture at room temperature for 12 h, the crude product was extracted. It was washed several times with ethyl acetate until it passed the AgNO_3_ test, then dried at approximately 100 °C to obtain the final product. The yield of the product was above 80%. ^1^H-NMR [400 MHz, D_2_O]: δ 0.80–0.89 (m, 18H, (*CH_3_*-CH_2_-)), 1.23–1.60 (m, 36H, (-*CH_2_-CH_2_*-CH_2_-P^+^-), (P^+^-CH_2_-*(CH_2_)_6_*-CH_2_-P^+^)), 2.10–2.23 (m, 16H, (-*CH_2_*-P^+^-)), 2.33−2.43 (s, 6H, *CH*_3_-Ph-), 7.31−7.38 (s, 4H, CH_3_-*PhH*-), 7.64−7.71 (s, 4H, CH_3_-*PhH*-).

#### 3.2.4. Dicationic Ammonium-Based ILs

In this study, octane-1,8-diylbis(tributylammonium) dibromide ([(N_4444_)_2_][Br]) was first obtained by stirring tributylamine (30.0 g, 160 mmol) and 1,8-dibromooctane (13.60 g, 50 mmol) in acetone (100 mL) in a glass bottle. The mixture was stirred at 100 °C for 48 h and then precipitated into 600 mL diethyl ether. The resulting solid was extracted and washed several times with ethyl acetate, then dried at approximately 100 °C to obtain the final product. The yield of the product was 8–10%. ^1^H-NMR [400 MHz, D_2_O]: δ 0.81–0.89 (m, 18H, (*CH_3_*-CH_2_-)), 1.22–1.38 (m, 20H, (-*CH_2_*-CH_2_-CH_2_-N^+^-), (N^+^-CH_2_-CH_2_ –*(CH_2_)_4_*-CH_2_-CH_2_-N^+^)), 1.48–1.68 (m, 16H, (-CH_2_*-CH_2_*-CH_2_-N^+^-), (N^+^-CH_2_-*CH_2_* –(CH_2_)_4_-*CH_2_*-CH_2_-N^+^)), 3.08–3.22 (m, 16H, (-CH_2_-CH_2_-*CH_2_*-N^+^-), (N^+^-*CH_2_*-(CH_2_)_6_-*CH_2_*-N^+^)).

Octane-1,8-diylbis(tributylammonium) benzenesulfonate ([(N_4444_)_2_][BS]) was obtained by stirring [(N_4444_)_2_][Br] (2.38 g, 3.70 mmol) and sodium benzenesulfonate (1.33 g, 7.40 mmol) in DI water (100 mL) in a glass bottle. After stirring the mixture at room temperature for 12 h, the crude product was extracted. It was washed several times with ethyl acetate until it passed the AgNO_3_ test, then dried at approximately 100 °C to obtain the final product. The yield of the product was 40–50%. ^1^H-NMR [400 MHz, D_2_O]: δ 0.83–1.00 (m, 18H, (*CH_3_*-CH_2_-)), 1.19–1.41 (m, 20H, (-*CH_2_*-CH_2_-CH_2_-N^+^-), (N^+^-CH_2_-CH_2_ –*(CH_2_)_4_*-CH_2_-CH_2_-N^+^)), 1.54–1.71 (m, 16H, (-CH_2_*-CH_2_*-CH_2_-N^+^-), (N^+^-CH_2_-*CH_2_* –(CH_2_)_4_-*CH_2_*-CH_2_-N^+^)), 3.11–3.26 (m, 16H, (-CH_2_-CH_2_-*CH_2_*-N^+^-), (N^+^-*CH_2_*-(CH_2_)_6_-*CH_2_*-N^+^)), 7.52−7.63 (s, 6H, *PhH*-), 7.78−7.83 (s, 4H, *PhH*-).

Octane-1,8-diylbis(tributylammonium) 4-methylbenzenesulfonate ([(N_4444_)_2_][MBS]) was obtained by stirring [(N_4444_)_2_][Br] (2.38 g, 3.70 mmol) and sodium 4-methylbenzenesulfonate (1.43 g, 7.40 mmol) in DI water (100 mL) in a glass bottle. After stirring the mixture at room temperature for 12 h, the crude product was extracted. It was washed several times with ethyl acetate until it passed the AgNO_3_ test, then dried at approximately 100 °C to obtain the final product. The yield of the product was 40–50%. ^1^H-NMR [400 MHz, D_2_O]: δ 0.88–0.98 (m, 18H, (*CH_3_*-CH_2_-)), 1.28–1.43 (m, 20H, (-*CH_2_*-CH_2_-CH_2_-N^+^-), (N^+^-CH_2_-CH_2_ –*(CH_2_)_4_*-CH_2_-CH_2_-N^+^)), 1.55–1.72 (m, 16H, (-CH_2_*-CH_2_*-CH_2_-N^+^-), (N^+^-CH_2_-*CH_2_* –(CH_2_)_4_-*CH_2_*-CH_2_-N^+^)), 2.36−2.43 (s, 6H, *CH*_3_-Ph-), 3.09–3.23 (m, 16H, (-CH_2_-CH_2_-*CH_2_*-N^+^-), (N^+^-*CH_2_*-(CH_2_)_6_-*CH_2_*-N^+^)), 7.32−7.39 (s, 4H, CH_3_-*PhH*-), 7.64−7.72 (s, 4H, CH_3_-*PhH*-).

### 3.3. Characterization

Proton nuclear magnetic resonance (^1^H-NMR) spectroscopy (MR400 DD2, Agilent Technologies, Inc., Santa Clara, CA, USA) and Fourier transform infrared (FT-IR) spectrometry with an attenuated total reflection accessory (NICOLET iS20, Thermo Fisher Scientific Inc., Waltham, MA, USA) were used to evaluate the structure of the synthesized ILs. The electrical conductivity of the sample was measured with a portable conductivity meter (Seven2Go pro, METTLER TOLEDO Inc., Columbus, OH, USA). The osmotic pressure of the sample was determined by measuring its freezing point using an osmometer using a plastic vial version (K-7400, KNAUER Wissenschaftliche Geräte GmbH Co., Berlin, Germany). The LCST was confirmed by measuring the transmittance of aqueous solutions with an ultraviolet-visible (UV-Vis) spectrophotometer (EMC-11D-V, EMCLAB Instruments GmbH Co., Duisburg, Germany) combined with a temperature controller (TC-200P, Misung Scientific. Co., Ltd., Yangju, Republic of Korea) [90]. The water flux was evaluated by measuring the difference in the volume of the draw solution in the tube at the beginning and end of the FO operation.

### 3.4. Forward Osmosis Tests

In water treatment technology, water flux is an important parameter that quantifies the movement of water across a membrane [91]. In simple terms, water flux is the rate at which water permeates across an FO membrane. The water flux was measured using a lab-scale FO system that connects two custom L-shaped glass tubes. A 3.325 × 10^−4^ m^2^ thin-film FO membrane (Hydration Technologies Inc., Albany, OR, USA) was placed in the channel between the two tubes. One side was filled with DI water as the feed solution, and the other contained the IL solution as the draw solution. The solutions were maintained at room temperature, with continued stirring. The water permeation flux (*J_v_*, L m^−2^ h^−1^ or LMH) was calculated from the volume of the draw solution at the beginning and end of the FO operation, as shown in Equation (2).
(2)$$J_{V}=\frac{\Delta V}{A\Delta t}$$
where ΔV (L) is the change in volume of the draw solution over time Δt (h) and A is the surface area of the membrane (m^2^).

## 4. Conclusions

A series of draw solutes with monocationic compounds ([P_4444_][BS], [P_4444_][MBS], [N_4444_][BS], and [N_4444_][MBS]) were obtained in a single step via anion exchange. Dicationic compounds ([(P_4444_)_2_][BS], [(P_4444_)_2_][MBS], [(N_4444_)_2_][BS], and [(N_4444_)_2_][MBS]) were prepared in two steps via the Menshutkin SN_2_ reaction and anion exchange. The resultant series of ILs, with various cations and anions, were applied as draw solutes in the FO process. We also investigated the suitability of ILs as draw solutes for FO systems based on their conductivity, osmotic pressure, and thermoresponsive behavior. The FO performance test was conducted at 5–20 wt%, and this concentration range is the concentration at which the eight draw solutes used in this study can be completely dissolved. Even within this range, the structural effect can be confirmed through the FO performance results. Conductivities and osmotic pressures trends of ILs were [(N_4444_)_2_][BS] > [(N_4444_)_2_][MBS] > [(P_4444_)_2_][BS] > [(P_4444_)_2_][MBS] > [N_4444_][BS] > [P_4444_][BS] > [N_4444_][MBS] > [P_4444_][MBS] at 5–20 wt%. The aqueous [P_4444_][BS], [N_4444_][BS], [N_4444_][MBS], and [(N_4444_)_2_][BS] solutions did not exhibit any thermal recovery properties (no LCST). However, 20 wt% aqueous [P_4444_][MBS], [(P_4444_)_2_][BS], [(P_4444_)_2_][MBS], and [(N_4444_)_2_][MBS] solutions were found to have LCSTs of approximately 43, 33, 22, and 60 °C, respectively. Moreover, the water flux of 20 wt% aqueous [(P_4444_)_2_][MBS] was measured to be approximately 10.58 LMH in the AL-DS mode and 9.40 LMH in the AL-FS mode. These results can be used to understand the structural effects of monocationic and dicationic ILs on LCST behavior and FO properties. Therefore, [(P_4444_)_2_][MBS] is the best candidate for draw solutes, owing to its relatively good FO performance and easy recovery.

## Figures and Tables

**Figure 1 molecules-27-08869-f001:**
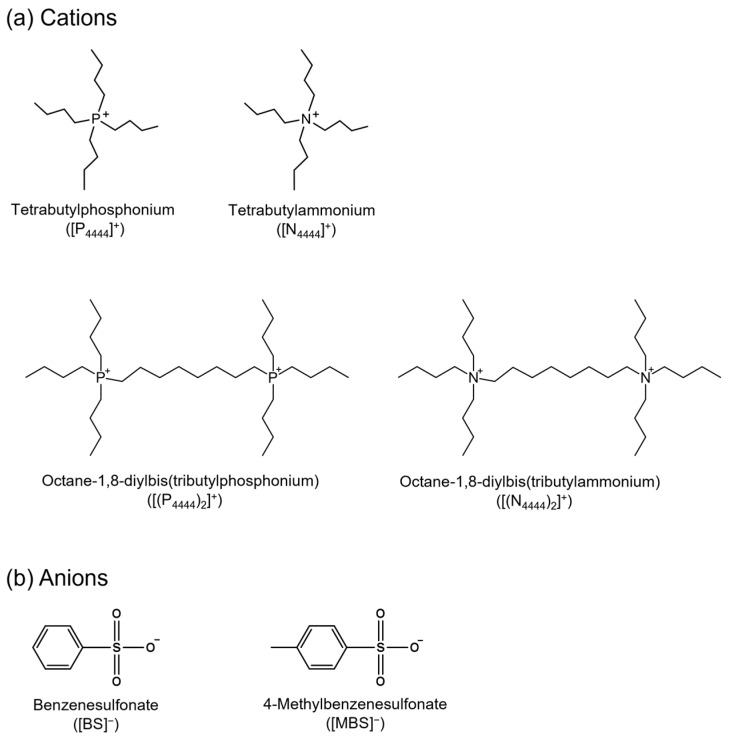
Chemical structures of (**a**) cations and (**b**) anions of the ionic liquid used in this study.

**Figure 2 molecules-27-08869-f002:**
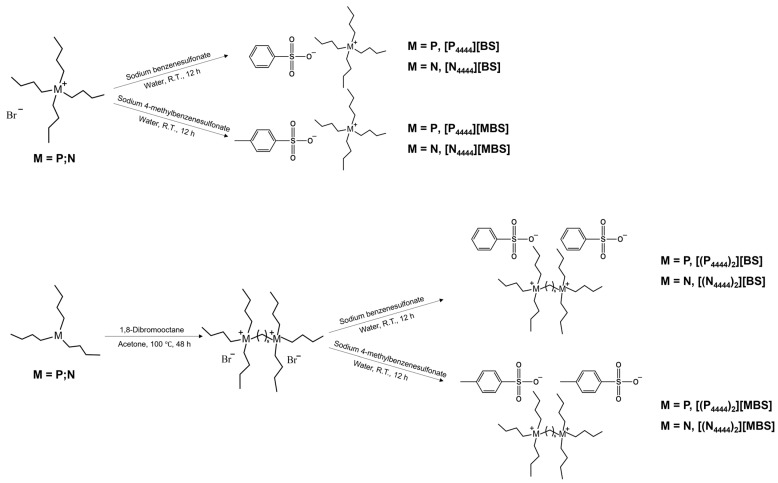
Synthetic route of the monocationic ILs (**top**) and dicationic ILs (**bottom**) used in this study.

**Figure 3 molecules-27-08869-f003:**
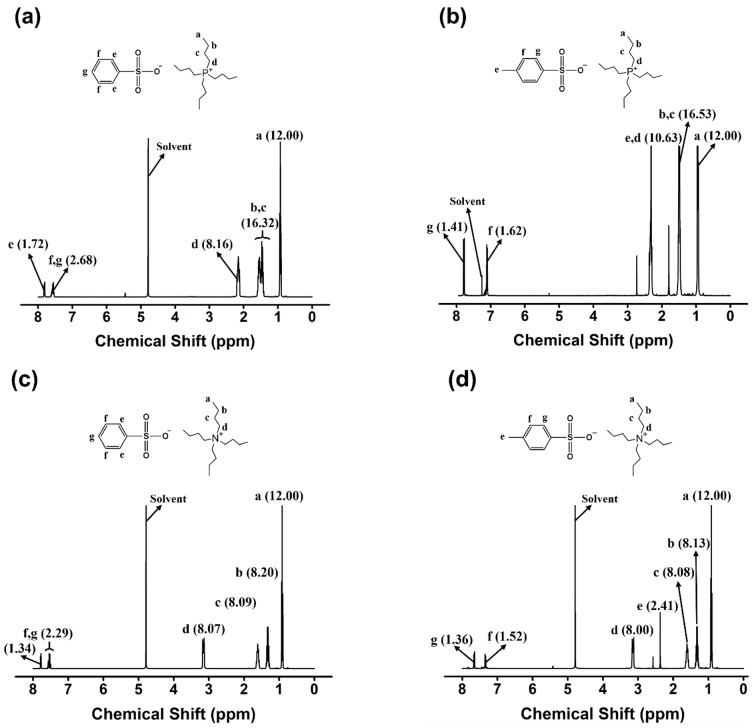
Proton nuclear magnetic resonance (^1^H NMR) spectrum of (**a**) [P_4444_][BS], (**b**) [P_4444_][MBS], (**c**) [N_4444_][BS], (**d**) [N_4444_][MBS], (**e**) [(P_4444_)_2_][BS], (**f**) [(P_4444_)_2_][MBS], (**g**) [(N_4444_)_2_][BS], and (**h**) [(N_4444_)_2_][MBS].

**Figure 4 molecules-27-08869-f004:**
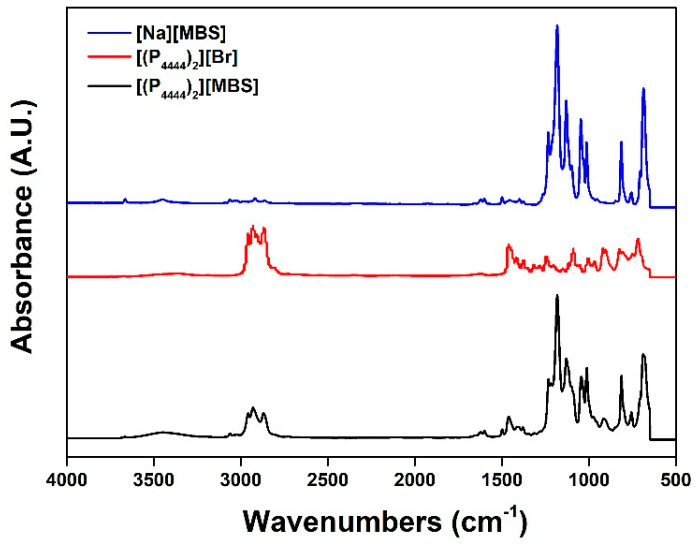
FT-IR spectra of aqueous of [Na][MBS], [(P_4444_)_2_][Br], and [(P_4444_)_2_][MBS] at room temperature.

**Figure 5 molecules-27-08869-f005:**
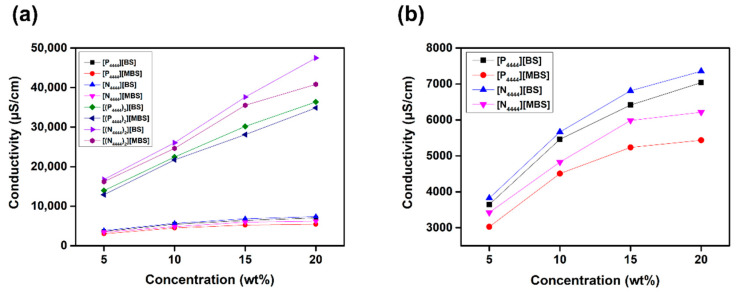
Conductivity of (**a**) [P_4444_][BS], [N_4444_][BS], [P_4444_][MBS], [N_4444_][MBS], [(P_4444_)_2_][BS], [(N_4444_)_2_][BS], [(P_4444_)_2_][MBS], and [(N_4444_)_2_][MBS] (**b**) [P_4444_][BS], [N_4444_][BS], [P_4444_][MBS], and [N_4444_][MBS] aqueous solutions according to concentration.

**Figure 6 molecules-27-08869-f006:**
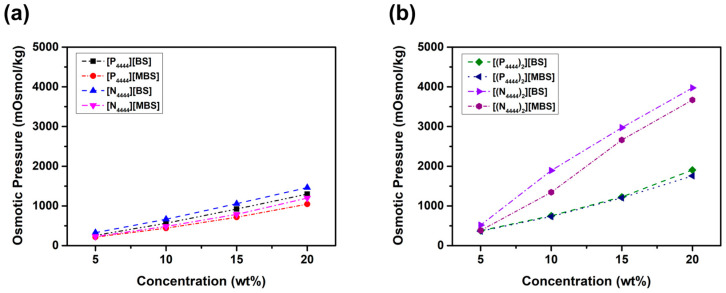
Osmotic pressure according to the concentration of (**a**) [P_4444_][BS], [N_4444_][BS], [P_4444_][MBS], and [N_4444_][MBS], (**b**) [(P_4444_)_2_][BS], [(N_4444_)_2_][BS], [(P_4444_)_2_][MBS], and [(N_4444_)_2_][MBS] measured by freezing point depression method.

**Figure 7 molecules-27-08869-f007:**
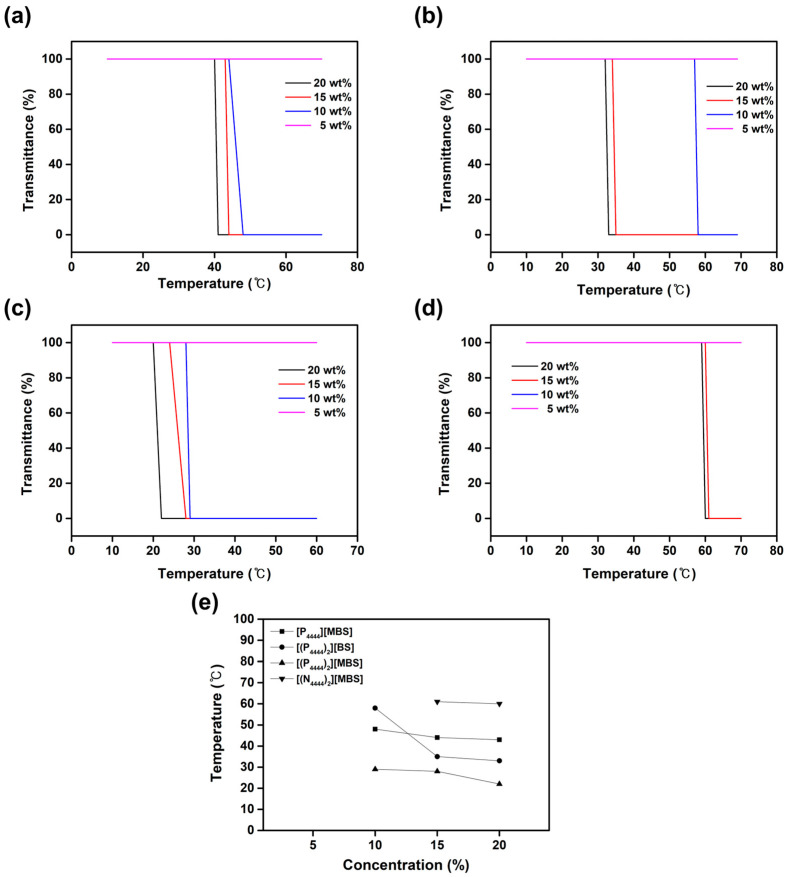
Transmittance curve of (**a**) [P_4444_][MBS], (**b**) [(P_4444_)_2_][BS], (**c**) [(P_4444_)_2_][MBS], and (**d**) [(N_4444_)_2_][MBS] according to the temperature change using UV-Vis spectrophotometer. (**e**) Phase diagram showing concentration dependence of LCST behavior of [P_4444_][MBS], [(P_4444_)_2_][BS], [(P_4444_)_2_][MBS], and [(N_4444_)_2_][MBS] solutions in water.

**Figure 8 molecules-27-08869-f008:**
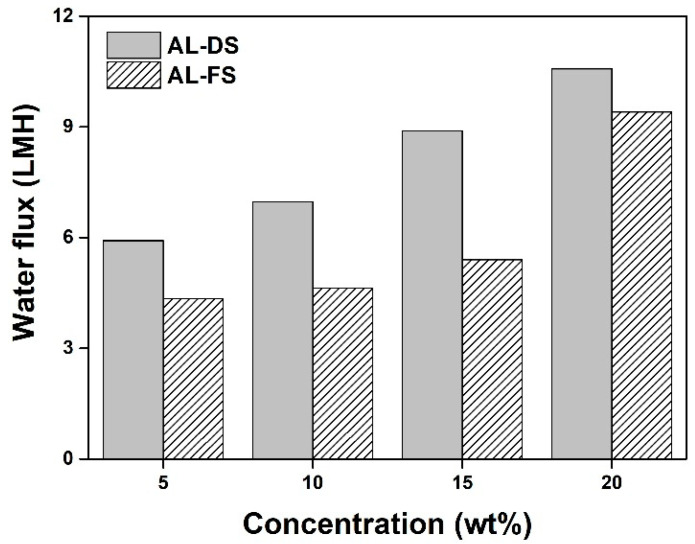
Water flux of octane-1,8-diylbis(tributylphosphonium) 2,4-dimethylbenzenesulfonate ([(P_4444_)_2_][MBS]) according to the concentration at room temperature during forward osmosis process in AL-DS mode and AL-FS mode.

**Figure 9 molecules-27-08869-f009:**
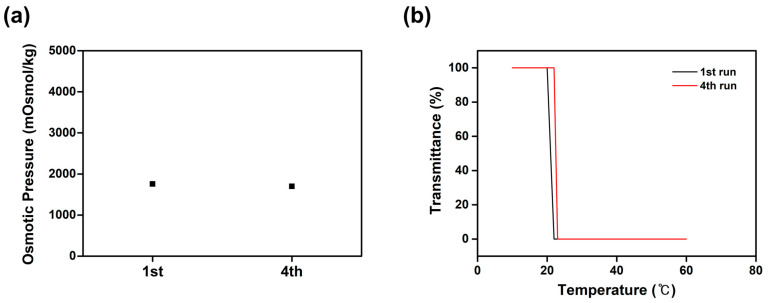
Recyclability study of [(P_4444_)_2_][MBS] in four cycles: (**a**) osmotic pressures and (**b**) thermoresponsive behavior tests. From the 2nd to the 4th run, the recovered [(P_4444_)_2_][MBS] from the previous run was used.

## Data Availability

The datasets are available from the corresponding author on reasonable request.

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
