# Peer review of "Thermoresponsive Ionic Liquid with Different Cation–Anion Pairs as Draw Solutes in Forward Osmosis"

_molecules, 2022, doi:10.3390/molecules27248869_

Round 1

Reviewer 1 Report

Manuscript is well written, and the experimental work is presented in an adequate manner. However, the authors have not included any proper phase diagrams showing the phase behavior of the ionic liquids with water. They may use the work reported in 'Separation and Purification Technology, 275 (2021) 119164' as a guide. All relevant phase diagrams should be included in the manuscript.

In the synthesis of ionic liquids, authors have used an anion exchange method to generate BS and MBS based ILs. Halide analysis of all ionic liquids has to be included in the experimental section.

No yields of ionic liquids can be found in the experimental section. All yields of synthesized ILs should be included.

There is less effort shown to regeneration of ionic liquids after osmosis experiments. At least a token method should be included in the manuscript.

Authors should revise their manuscript to accommodate all the above concerns and suggestions. 

Author Response

Dear Editor at Molecules

We gratefully appreciate your kind review and consideration for publication in “Molecules”. We are submitting a revised manuscript (molecules-2066491) entitled “Thermoresponsive ionic liquid with different cation-anion pairs as draw solutes in forward osmosis”.

We carefully read the reviewer’s comments and your e-mail. Reviewers gave us helpful comments for our manuscript. We think the reviewer’s opinion and suggestion are fairly reasonable. Therefore, we revised our manuscript taking the reviewer’s comments into consideration as follows. As you and the Reviewer suggested we modified some parts of the manuscript, and the changes are shown as yellow texts. These changes are listed as follows:

Referee’s comments:

Referee: 1

Comments:
Manuscript is well written, and the experimental work is presented in an adequate manner.

- Main comments: However, the authors have not included any proper phase diagrams showing the phase behavior of the ionic liquids with water. They may use the work reported in 'Separation and Purification Technology, 275 (2021) 119164' as a guide. All relevant phase diagrams should be included in the manuscript.

Answer:

We are much obliged for comment suggested by Reviewer, and we added the phase diagram data in the into new Figure 7 (e) as shown below and added the text as “They exhibited a typical LCST-type phase separation behavior by changing the IL concentration as shown in their phase diagrams (Fig. 7(e)).” in the Results and Discussion section.

Figure 7. Transmittance curve of (a) [P4444][MBS], (b) [(P4444)2][BS], (c) [(P4444)2][MBS], and (d) [(N4444)2][MBS] according to the temperature change using UV-Vis spectrophotometer. (e) Phase diagram showing concentration dependence of LCST behavior of [P4444][MBS], [(P4444)2][BS], [(P4444)2][MBS], and [(N4444)2][MBS] solutions in water.

- In the synthesis of ionic liquids, authors have used an anion exchange method to generate BS and MBS based ILs. Halide analysis of all ionic liquids has to be included in the experimental section.

Answer:

We deeply thank for the comments suggested by Reviewer, and we added the text as “until it passed the AgNO3 test, commonly known as halide analysis of IL [89].” and “until it passed the AgNO3 test.” in the Experimental section. And added reference number 89 to the References section. Added literatures are below.

  1. Wu, B.; Yamashita, Y.; Endo, T.; Takahashi, K.; Castner Jr, E.W. Structure and Dynamics of Ionic Liquids: Trimethylsilylpropyl-Substituted Cations and Bis (Sulfonyl) Amide Anions. Chem. Phys.2016, 145, 244506. https://doi.org/10.1063/1.4972410.

- No yields of ionic liquids can be found in the experimental section. All yields of synthesized ILs should be included.

Answer:

We deeply thank for the comments suggested by Reviewer, and we added the text as “The yield of the product was above 80%.” and “The yield of the product was 40–50%.” and “The yield of the product was 8–10%.” in the Experimental section.

- There is less effort shown to regeneration of ionic liquids after osmosis experiments. At least a token method should be included in the manuscript.

Answer:

We are much obliged for comment suggested by Reviewer, and we added the recyclability data of [(P4444)2][MBS] in the into new Figure 9 as shown below and added the text as “To demonstrate the recyclability of [(P4444)2][MBS] after osmotic experiments, FO process was repeated four times using a 20 wt% solution of [(P4444)2][MBS] as the draw solution and DI water as the feed solution. When the temperature rises above the critical temperature after the permeation process, [(P4444)2][MBS] is precipitated in the solution, and pure water can be easily separated by a simple filtration process. As shown in Fig. 9, to confirm the recyclability of [(P4444)2][MBS], the osmotic pressure and thermoresponsive behavior tests of [(P4444)2][MBS] were measured at the 4th run, respectively. Osmotic pressure and LCST value of [(P4444)2][MBS] in 4 runs are almost similar to the pristine [(P4444)2][MBS]. These recycling results clearly show that [(P4444)2][MBS] can be easily recycled with relatively low energy consumption without significant loss.” in the Results and Discussion section. We strongly believe that Reviewer will satisfy the answer to this question.

Figure 9. Recyclability study of [(P4444)2][MBS] in four cycles: (a) osmotic pressures and (b) thermoresponsive behavior tests. From the 2nd to the 4th run, the recovered [(P4444)2][MBS] from the previous run was used.

We believe that now we answered all of the comments pointed out by the reviewers. I hope that now this paper is publishable in “Molecules”, one of the top journals in molecular science area.

We also believe that this paper is also suitable for publication in “Molecules” from the following reasons.

  1. We synthesized various phosphonium- and ammonium-based ionic liquids (ILs), using benzenesulfonate (BS) and 4-methylbenzenesulfonate (MBS) to establish the criteria for designing an ideal draw solute in a forward osmosis (FO) system. Additionally, the effects of monocationic, dicationic, and anionic species on FO performance were studied. Monocationic compounds ([P4444][BS], [P4444][MBS], [N4444][BS], and [N4444][MBS]) were obtained in one step via anion exchange. Dicationic compounds ([(P4444)2][BS], [(P4444)2][MBS], [(N4444)2][BS], and [(N4444)2][MBS]) were prepared in two steps via a Menshutkin SN2 reaction and anion exchange. We also investigated the suitability of ILs as draw solutes for FO systems based on their thermoresponsive behavior, conductivity, and osmotic pressure.

  1. The aqueous [P4444][BS], [N4444][BS], [N4444][MBS], and [(N4444)2][BS] solutions did not exhibit thermoresponsive behavior. However, 20 wt% [P4444][MBS], [(P4444)2][BS], [(P4444)2][MBS], and [(N4444)2][MBS] had critical temperatures of approximately 43, 33, 22, and 60 °C, respectively, enabling their recovery using temperature. Moreover, the FO performance of 20 wt% aqueous [(P4444)2][MBS] solution was tested for water flux and found to be approximately 10.58 L m-2 h-1 (LMH) with the active layer facing the draw solution (AL-DS) mode and 9.40 LMH with the active layer facing the feed solution (AL-FS).

  1. These results can be used to understand the structural effects of monocationic and dicationic ILs on LCST behavior and FO properties. Therefore, [(P4444)2][MBS] is the best candidate for draw solutes, owing to its relatively good FO performance and easy recovery.

Thank you very much for your time and consideration on this process. I am looking forward to hearing good news from you very soon.

Sincerely (on behalf of all authors),

Prof. Hyo Kang

Associate Professor
Department of Chemical Engineering
Dong-A University
Busan 49315, Republic of Korea
Tel: +82 51 200 7720
Fax: +82 51 200 7728
E-mail hkang@dau.ac.kr

Reviewer 2 Report

The authors report interesting work on the synthesis of phosphonium/ammonium- based monocationic/dicationic ILs and their uses as draw solutes in FO. The results indicate that 20 wt% (P444)2(MBS) has a LCST closest to room temperature and is the most promising draw solution. I feel that the manuscript is well-structured and would recommend publication after minor revisions. My detailed comments are as follows:

1. Abstract: It would be better to include some mechanisms and discussions of the results.

2. The synthetic paths are well-established. Please add relevant reference. Please also add all peak areas in the NMR spectrums, since they are critical factors for the molecular structures.

3. Is there a specific objective of using benzenesulfonate-based anions?

4. Figures 5 and 6: Please use the same y-axis scales for a and b.

5. Why did the authors only study IL concentrations <20 wt%? How would the conductivities, osmotic pressures and thermoresponses behave for IL solutions >20 wt%? The results seem to show higher concentration has better performances.

6. Is there a reference for using UV-Vis to determine LCST? Why does transmittance decrease so much after separation occurs? The authors also hypothesize that hydrogen bond with water is stronger than ion-ion interaction below LCST and weaker above LCST. Is there a fundamental background for such hypothesis?

7. How would the water flux be for (P444)2(MBS) compared to other candidates? (P444)2(MBS) shows not the highest conductivity and osmotic pressure.

Author Response

Dear Editor at Molecules

We gratefully appreciate your kind review and consideration for publication in “Molecules”. We are submitting a revised manuscript (molecules-2066491) entitled “Thermoresponsive ionic liquid with different cation-anion pairs as draw solutes in forward osmosis”.

We carefully read the reviewer’s comments and your e-mail. Reviewers gave us helpful comments for our manuscript. We think the reviewer’s opinion and suggestion are fairly reasonable. Therefore, we revised our manuscript taking the reviewer’s comments into consideration as follows. As you and the Reviewer suggested we modified some parts of the manuscript, and the changes are shown as yellow texts. These changes are listed as follows:

Referee’s comments:

Referee: 2

Comments:
The authors report interesting work on the synthesis of phosphonium/ammonium- based monocationic/dicationic ILs and their uses as draw solutes in FO. The results indicate that 20 wt% (P4444)2(MBS) has a LCST closest to room temperature and is the most promising draw solution. I feel that the manuscript is well-structured and would recommend publication after minor revisions. My detailed comments are as follows:

- Abstract: It would be better to include some mechanisms and discussions of the results.

Answer:

We deeply thank for the comments suggested by Reviewer, and we amended the text as “We synthesized various phosphonium- and ammonium-based ionic liquids (ILs), using benzenesulfonate (BS) and 4-methylbenzenesulfonate (MBS) to establish the criteria for designing an ideal draw solute in a forward osmosis (FO) system. Additionally, the effects of monocationic, dicationic, and anionic species on FO performance were studied. Monocationic compounds ([P4444][BS], [P4444][MBS], [N4444][BS], and [N4444][MBS]) were obtained in one step via anion exchange. Dicationic compounds ([(P4444)2][BS], [(P4444)2][MBS], [(N4444)2][BS], and [(N4444)2][MBS]) were prepared in two steps via a Menshutkin SN2 reaction and anion exchange. We also investigated the suitability of ILs as draw solutes for FO systems. The aqueous [P4444][BS], [N4444][BS], [N4444][MBS], and [(N4444)2][BS] solutions did not exhibit thermoresponsive behavior. However, 20 wt% [P4444][MBS], [(P4444)2][BS], [(P4444)2][MBS], and [(N4444)2][MBS] had critical temperatures of approximately 43, 33, 22, and 60 °C, respectively, enabling their recovery using temperature. An increase in IL hydrophobicity and bulkiness reduces its miscibility with water, demonstrating that it can be used to tune its thermoresponsive properties. Moreover, the FO performance of 20 wt% aqueous [(P4444)2][MBS] solution was tested for water flux and found to be approximately 10.58 LMH with the active layer facing the draw solution mode and 9.40 LMH with the active layer facing the feed solution.” in the Abstract section.

- The synthetic paths are well-established. Please add relevant reference. Please also add all peak areas in the NMR spectrums, since they are critical factors for the molecular structures.

Answer:

We deeply thanks for comments suggested by Reviewer and we added the text as “using modified literature procedures [60–62].” and added the literature in the References section as number of 60–62, respectively. Added literatures are below. And we added all peak areas in the NMR spectrums as shown below in Results and discussion section.

  1. Kim, T.; Ju, C.; Park, C.; Kang, H. Polymer having Dicationic Structure in Dumbbell Shape for Forward Osmosis Process. Polymers 2019, 11, 571. https://doi.org/10.3390/polym11030571.
  2. Men, Y.; Schlaad, H.; Voelkel, A.; Yuan, J. Thermoresponsive Polymerized Gemini Dicationic Ionic Liquid. Chem. 2014, 5, 3719–3724. https://doi.org/10.1039/C3PY01790G.
  3. Liu, P.; Wang, D.C.; Ho, C.; Chen, Y.; Chung, L.; Liang, T.; Chang, M.; Horng, R. Exploring the Performance-Affecting Factors of Monocationic and Dicationic Phosphonium-Based Thermoresponsive Ionic Liquid Draw Solutes in Forward Osmosis. Desalination Water Treat. 2020, 200, 1–7. 10.5004/dwt.2020.25987.

Figure 3. Proton nuclear magnetic resonance (1H NMR) spectrum of (a) [P4444][BS], (b) [P4444][MBS], (c) [N4444][BS], (d) [N4444][MBS], (e) [(P4444)2][BS], (f) [(P4444)2][MBS], (g) [(N4444)2][BS], and (h) [(N4444)2][MBS].

- Is there a specific objective of using benzenesulfonate-based anions?

Answer:

We deeply thank for the comments suggested by Reviewer, and we added the text as “This indicates that properly designed monocationic-based ILs and dicationic-based ILs achieve the desired physicochemical and biological properties. The selection of an appropriate counter ion is also very important for IL characterization [42]. Representative anions of ILs include sulfonate, carboxylate, phosphate, halides, and amino acids [43–45]. These anions are used to control water miscibility by altering the alkyl chain length or by attaching non-polar groups to maintain a hydrophilic/hydrophobic balance [46–48]. For this reason, it is necessary to pay attention to benzenesulfonate, which is a form in which phenyl, a simple hydrophobic group, is attached to a sulfonate group. According to previous literature, benzenesulfonate is the simplest member of the class of a benzenesulfonic acids, with high thermal stability and low manufacturing costs [49,50].” in the Introduction section and added the literature in the References section as number of 42–50, respectively. Added literatures are below.

  1. Kaczmarek, D.K.; Gwiazdowska, D.; Marchwińska, K.; Klejdysz, T.; Wojcieszak, M.; Materna, K.; Pernak, J. Amino Acid-Based Dicationic Ionic Liquids as Complex Crop Protection Agents. Mol. Liq. 2022, 360, 119357. https://doi.org/10.1016/j.molliq.2022.119357.
  2. Zhou, J.; Sui, H.; Jia, Z.; Yang, Z.; He, L.; Li, X. Recovery and Purification of Ionic Liquids from Solutions: A Review. RSC Adv. 2018, 8, 32832–32864. http://dx.doi.org/10.1039/C8RA06384B.
  3. Correia, D.M.; Fernandes, L.C.; Fernandes, M.M.; Hermenegildo, B.; Meira, R.M.; Ribeiro, C.; Ribeiro, S.; Reguera, J.; Lanceros-Méndez, S. Ionic Liquid-Based Materials for Biomedical Applications. Nanomaterials2021, 11, 2401. https://doi.org/10.3390/nano11092401.
  4. Rajyaguru, Y.V.; Patil, J.H.; Kusanur, R. Ionic Liquids, an Asset in Extraction Techniques–a Comprehensive Review. and Adv. in Chem. 2022, 12, 107–122. https://doi.org/10.1134/S2634827622020040.
  5. Huddleston, J.G.; Visser, A.E.; Reichert, W.M.; Willauer, H.D.; Broker, G.A.; Rogers, R.D. Characterization and Comparison of Hydrophilic and Hydrophobic Room Temperature Ionic Liquids Incorporating the Imidazolium Cation. Green Chem.2001, 3, 156–164. https://doi.org/10.1039/B103275P.
  6. Schröder, C. Proteins in Ionic Liquids: Current Status of Experiments and Simulations. Ionic liquids II2017, 127–152. https://doi.org/10.1007/s41061-017-0110-2.
  7. Nikfarjam, N.; Ghomi, M.; Agarwal, T.; Hassanpour, M.; Sharifi, E.; Khorsandi, D.; Ali Khan, M.; Rossi, F.; Rossetti, A.; Nazarzadeh Zare, E. Antimicrobial Ionic Liquid‐based Materials for Biomedical Applications. Funct. Mater. 2021, 31, 2104148. https://doi.org/10.1002/adfm.202104148.
  8. Thomas, S.; Rayaroth, M.P.; Menacherry, S.P.M.; Aravind, U.K.; Aravindakumar, C.T. Sonochemical Degradation of Benzenesulfonic Acid in Aqueous Medium. Chemosphere 2020, 252, 126485. https://doi.org/10.1016/j.chemosphere.2020.126485
  9. Masoudian, Z.; Salehi-Lisar, S.Y.; Norastehnia, A. Phytoremediation Potential of Azolla Filiculoides for Sodium Dodecyl Benzene Sulfonate (SDBS) Surfactant Considering some Physiological Responses, Effects of Operational Parameters and Biodegradation of Surfactant. Sci. Pollut. Res. 2020, 27, 20358–20369. https://doi.org/10.1007/s11356-020-08286-2.

- Figures 5 and 6: Please use the same y-axis scales for a and b.

Answer:

We are much obliged for comment suggested by Reviewer, and we revised Figure 5 and 6 of previous version manuscript in the Results and Discussion section as below.

Figure 5. Conductivity of (a) [P4444][BS], [N4444][BS], [P4444][MBS], [N4444][MBS], [(P4444)2][BS], [(N4444)2][BS], [(P4444)2][MBS], and [(N4444)2][MBS] (b) [P4444][BS], [N4444][BS], [P4444][MBS], and [N4444][MBS] aqueous solutions according to concentration.

Figure 6. Osmotic pressure according to the concentration of (a) [P4444][BS], [N4444][BS], [P4444][MBS], and [N4444][MBS], (b) [(P4444)2][BS], [(N4444)2][BS], [(P4444)2][MBS], and [(N4444)2][MBS] measured by freezing point depression method.

- Why did the authors only study IL concentrations <20 wt%? How would the conductivities, osmotic pressures and thermoresponses behave for IL solutions >20 wt%? The results seem to show higher concentration has better performances.

Answer:

We deeply thank for the comments suggested by Reviewer, and we added the text as “However, it is known that in a high concentration range outside of a concentration range in which conductivity and concentration are proportional, the interaction force between ions is strengthened and the mobility of ions is weakened, thereby decreasing the conductivity of the aqueous solution [64].” and “That is, according to Equation 1, the proportional relationship between the osmotic pressure and the concentration can be applied even to the highest concentration at which the draw solute can be dissolved in water.” and “However, it is known that a reversed Tcloud/concentration relationship is observed at much higher draw solution concentrations [81–83].” in the Results and discussion section and added the literature in the References section as number of 64, 81–83 respectively. Added literatures are below. And we added the text as “We also investigated the suitability of ILs as draw solutes for FO systems based on their conductivity, osmotic pressure, and thermoresponsive behavior. The FO performance test was conducted at 5–20 wt%, and this concentration range is the concentration at which the eight draw solutes used in this study can be completely dissolved. Even within this range, the structural effect can be confirmed through the FO performance results. Conductivities and osmotic pressures trends of ILs were [(N4444)2][BS] > [(N4444)2][MBS] > [(P4444)2][BS] > [(P4444)2][MBS] > [N4444][BS] > [P4444][BS] > [N4444][MBS] > [P4444][MBS] at 5–20 wt%.” in the Conclusions section.

  1. Zhang, Z.; Madsen, L.A. Observation of Separate Cation and Anion Electrophoretic Mobilities in Pure Ionic Liquids. Chem. Phys. 2014, 140, 084204. https://doi.org/10.1063/1.4865834.
  2. Dong, S.; Heyda, J.; Yuan, J.; Schalley, C.A. Lower Critical Solution Temperature (LCST) Phase Behaviour of an Ionic Liquid and its Control by Supramolecular Host–guest Interactions. Commun. 2016, 52, 7970–7973. https://doi.org/10.1039/C6CC02838A.
  3. Zhang, J.; Jiang, X.; Wen, X.; Xu, Q.; Zeng, H.; Zhao, Y.; Liu, M.; Wang, Z.; Hu, X.; Wang, Y. Bio-Responsive Smart Polymers and Biomedical Applications. Phys. Materials2019, 2, 032004. http://dx.doi.org/10.1088/2515-7639/ab1af5.
  4. Sugeno, K.; Kokubun, S.; Saito, H. Ucst Type Phase Boundary and Accelerated Crystallization in Ptt/Pet Blends. Polymers 2020, 12, 2730. http://dx.doi.org/10.3390/polym12112730.

- Is there a reference for using UV-Vis to determine LCST? Why does transmittance decrease so much after separation occurs? The authors also hypothesize that hydrogen bond with water is stronger than ion-ion interaction below LCST and weaker above LCST. Is there a fundamental background for such hypothesis?

Answer:

We deeply thanks for comments suggested by Reviewer and we added the text as “When the temperature decreases below the critical temperature of the draw solution, molecular motion decreases, forming hydrogen bonds between the water molecules and the ionic groups of the draw solute, resulting in complete mixing [75]. Therefore, below the LCST, the transmittance of the aqueous solution is close to 100% because the draw solution is optically transparent and homogeneous. As the temperature increases above the critical temperature, the increase in molecular motion breaks the hydrogen bonds between the water molecules and the ionic groups of the draw solute, resulting in IL aggregation as ion-ion interactions become dominant [76,77]. Therefore, above the LCST, the aqueous solution becomes cloudy and blocks light, so the transmittance of the aqueous solution was approximately 0% [78].” in Results and discussion section. and added the literature in the References section as number of 75–78, 90 respectively. Added literatures are below.

  1. Itsuki, K.; Kawata, Y.; Sharker, K.K.; Yusa, S. Ultrasound-and Thermo-Responsive Ionic Liquid Polymers. Polymers2018, 10, 301. https://doi.org/10.3390/polym10030301.
  2. Nash, M.E.; Carroll, W.M.; Velasco, D.; Gomez, J.; Gorelov, A.V.; Elezov, D.; Gallardo, A.; Rochev, Y.A.; Elvira, C. Synthesis and Characterization of a Novel Thermoresponsive Copolymer Series and their Application in Cell and Cell Sheet Regeneration. Biomater. Sci. Polym. Ed. 2013, 24, 253–268. https://doi.org/10.1080/09205063.2012.690274.
  3. Zhao, Y.; Wang, H.; Pei, Y.; Liu, Z.; Wang, J. Understanding the Mechanism of LCST Phase Separation of Mixed Ionic Liquids in Water by MD Simulations. Chem. Chem. Phys. 2016, 18, 23238–23245. https://doi.org/10.1039/C6CP03439J.
  4. Idziak, I.; Avoce, D.; Lessard, D.; Gravel, D.; Zhu, X.X. Thermosensitivity of Aqueous Solutions of Poly (N, N-Diethylacrylamide). Macromolecules1999, 32, 1260–1263. https://doi.org/10.1021/ma981171f.
  5. Idziak, I.; Avoce, D.; Lessard, D.; Gravel, D.; Zhu, X.X. Thermosensitivity of Aqueous Solutions of Poly (N,N-Diethylacrylamide). Macromolecules 1999, 32, 1260–1263. https://doi.org/10.1021/ma981171f.

- How would the water flux be for (P4444)2(MBS) compared to other candidates? (P4444)2(MBS) shows not the highest conductivity and osmotic pressure.

Answer:

We are much obliged for comment suggested by Reviewer, and we added the text as “The higher osmotic pressure of the draw solution than that of feed solution can create a driving force for water transport from the feed solution, resulting in a higher water flux to the draw solution in the FO process [87]. Thus, the order of the flux results for the eight draw solutes used in this study is expected to be the same as the previous osmotic pressure results (Fig. 6). Based on previous experimental results, among these eight draw solutes, [(P4444)2][MBS] was chosen as a representative IL because it exhibits relatively good FO performance and the most energy efficient LCST.” in Results and discussion section and added reference number 87 to the References section. Added literatures are below. We strongly believe that Reviewer will satisfy the answer to this question. And

  1. McCutcheon, J.R.; McGinnis, R.L.; Elimelech, M. Desalination by Ammonia–carbon Dioxide Forward Osmosis: Influence of Draw and Feed Solution Concentrations on Process Performance. Membr. Sci.2006, 278, 114–123. https://doi.org/10.1016/j.memsci.2005.10.048.

We believe that now we answered all of the comments pointed out by the reviewers. I hope that now this paper is publishable in “Molecules”, one of the top journals in molecular science area.

We also believe that this paper is also suitable for publication in “Molecules” from the following reasons.

  1. We synthesized various phosphonium- and ammonium-based ionic liquids (ILs), using benzenesulfonate (BS) and 4-methylbenzenesulfonate (MBS) to establish the criteria for designing an ideal draw solute in a forward osmosis (FO) system. Additionally, the effects of monocationic, dicationic, and anionic species on FO performance were studied. Monocationic compounds ([P4444][BS], [P4444][MBS], [N4444][BS], and [N4444][MBS]) were obtained in one step via anion exchange. Dicationic compounds ([(P4444)2][BS], [(P4444)2][MBS], [(N4444)2][BS], and [(N4444)2][MBS]) were prepared in two steps via a Menshutkin SN2 reaction and anion exchange. We also investigated the suitability of ILs as draw solutes for FO systems based on their thermoresponsive behavior, conductivity, and osmotic pressure.

  1. The aqueous [P4444][BS], [N4444][BS], [N4444][MBS], and [(N4444)2][BS] solutions did not exhibit thermoresponsive behavior. However, 20 wt% [P4444][MBS], [(P4444)2][BS], [(P4444)2][MBS], and [(N4444)2][MBS] had critical temperatures of approximately 43, 33, 22, and 60 °C, respectively, enabling their recovery using temperature. Moreover, the FO performance of 20 wt% aqueous [(P4444)2][MBS] solution was tested for water flux and found to be approximately 10.58 L m-2 h-1 (LMH) with the active layer facing the draw solution (AL-DS) mode and 9.40 LMH with the active layer facing the feed solution (AL-FS).

  1. These results can be used to understand the structural effects of monocationic and dicationic ILs on LCST behavior and FO properties. Therefore, [(P4444)2][MBS] is the best candidate for draw solutes, owing to its relatively good FO performance and easy recovery.

Thank you very much for your time and consideration on this process. I am looking forward to hearing good news from you very soon.

Sincerely (on behalf of all authors),

Prof. Hyo Kang

Associate Professor
Department of Chemical Engineering
Dong-A University
Busan 49315, Republic of Korea
Tel: +82 51 200 7720
Fax: +82 51 200 7728
E-mail hkang@dau.ac.kr

Round 2

Reviewer 1 Report

I am satisfied with the changes made to the manuscript by the authors. However, the authors have to realise  one could produce better quality phase diagrams and also AgNO3 test for chloride detection is very qualitative. I am happy to overlook these factors and allow its publication in its current form.